# Conductive Polymers and Their Nanocomposites as Adsorbents in Environmental Applications

**DOI:** 10.3390/polym13213810

**Published:** 2021-11-04

**Authors:** Mohammad Ilyas Khan, Mohammed Khaloufa Almesfer, Abubakr Elkhaleefa, Ihab Shigidi, Mohammed Zubair Shamim, Ismat H. Ali, Mohammad Rehan

**Affiliations:** 1Department of Chemical Engineering, College of Engineering, King Khalid University, Abha 62529, Saudi Arabia; almesfer@kku.edu.sa (M.K.A.); amelkhalee@kku.edu.sa (A.E.); etaha@kku.edu.sa (I.S.); 2Department of Electrical Engineering, College of Engineering, King Khalid University, Abha 62529, Saudi Arabia; mzmohammad@kku.edu.sa; 3Department of Chemistry, College of Science, King Khalid University, Abha 62529, Saudi Arabia; ismathassanali@gmail.com; 4Centre of Excellence in Environmental Studies, King Abdulaziz University, Jeddah 21577, Saudi Arabia; dr.mohammad_rehan@yahoo.co.uk

**Keywords:** conductive polymers, nanocomposites, adsorption, environmental remediation, heavy metal ions, organic dyes, gaseous pollutants

## Abstract

Proper treatment and disposal of industrial pollutants of all kinds are a global issue that presents significant techno-economical challenges. The presence of pollutants such as heavy metal ions (HMIs) and organic dyes (ODs) in wastewater is considered a significant problem owing to their carcinogenic and toxic nature. Additionally, industrial gaseous pollutants (GPs) are considered to be harmful to human health and may cause various environmental issues such as global warming, acid rain, smog and air pollution, etc. Conductive polymer-based nanomaterials have gained significant interest in recent years, compared with ceramics and metal-based nanomaterials. The objective of this review is to provide detailed insights into different conductive polymers (CPs) and their nanocomposites that are used as adsorbents for environmental remediation applications. The dominant types of CPs that are being used as adsorbent materials include polyaniline (PANI), polypyrrole (Ppy), and polythiophene (PTh). The various adsorption mechanisms proposed for the removal of ODs, HMIs, and other GPs by the different CPs are presented, together with their maximum adsorption capacities, experimental conditions, adsorption, and kinetic models reported.

## 1. Introduction

Industrialization, human development, and socio-economic activities are known to have led to the destruction and deterioration of the environment and have significantly affected human health. For example, the release of many different types of organic, inorganic, and gaseous pollutants into the environment and their subsequent penetration and accumulation in the food chain can be a significant threat to the environment and to human health [1]. It is well known that quite a significant portion of diseases and deaths are caused by consuming contaminated water and other polluted environmental issues in developing countries [2]. Among the inorganic contaminants, heavy metal ions (HMIs) are known to be the primary environmental contaminants [3]. The main sources of HMIs are metal processing and finishing industries [4], battery and electroplating industries, tanneries, glass and ceramic industries, as well as petroleum refining and mining industries [5,6]. Organic pollutants (OPs) vary in nature and have different environmental effects. Some organic pollutants are considered more toxic and harmful, compared with others. Organic dyes (ODs), primarily from textile industries, paper and pulp manufacturing, leather processing, food processing, pharmaceutical, and paints and coatings industries, are the main sources of contamination of wastewater [7,8]. Among gaseous pollutants (GPs), carbon dioxide (CO_2_), also known as the greenhouse gas, is known to be associated with many industries such as power plants [9], petrochemical industries, hydrogen, and cement manufacturing plants [10], and is suspected to be the primary pollutant responsible for ozone layer degradation and subsequently for global warming.

In order to minimize the number of such pollutants and to meet the environmental standards and regulations, various treatment techniques have been reported in the literature, which are summarized in Figure 1. Among these, adsorption is known to be more practical, adaptable, feasible, efficient, and environmentally friendly, compared with the other available techniques.

Conductive polymers (CPs) are well known for their outstanding characteristics. CPs consist of conjugated π-bonds and offer unique electrical, optical, and physical properties. Some of the most common types of conductive polymers are listed in Table 1 [11]. Conductive polymers and their nanocomposites have gained tremendous popularity in recent years as being widely used in diverse fields of research and innovations. Their potential use as biosensors [12,13], gas sensors [14], and corrosion inhibitors in different environments [15,16,17], as well as in biomedical applications [18,19,20,21], or as adsorbents for the adsorption of various environmental pollutants [22], is widely anticipated and frequently reported in the literature.

There is no single and comprehensive review that is solely dedicated to the applications and utilization of conductive polymers for environmental remediation. The only review article that provides deeper insights into the useful applications of conductive polymers is presented by [22], which contains a section that deals with environmental remediation but is not exclusive. Aside from this review article, another review by [23] has highlighted the application of conductive polymers in water treatment. There are various other reviews but are mostly concerned with specific conductive polymer and its composites. The objective of the current review is to combine recent developments of different conductive polymers (such as polypyrrole, polyaniline, and polythiophene) and their nanocomposites in environmental remediation in one comprehensive article. This review also sheds light on the various reported aspects of CPs as adsorbents for heavy metal removal, maximum adsorption capacities, adsorption mechanisms, adsorption, and kinetic models.

## 2. Conductive Polymers as Adsorbents

Conductive polymers and their nanocomposites as adsorbent materials have been shown to be effective and efficient in environmental remediation applications. This is most likely due to their interesting redox characteristics especially PANI [24], as well as other physical and chemical properties. The sorption characteristics of these adsorbents strongly depend on the solution pH, initial concentration, contact time, adsorbent dosage, and temperature, as well as on the operating pressure in the case of gaseous pollutants removal. Section 2.1 of this article deals with polyaniline and its composites adsorbents for heavy metal ions, organic and pharmaceutical pollutants, organic dyes, and gaseous pollutants removal. Section 2.2 is dedicated to polypyrrole and its composites for heavy metal ions, organic dyes, and gaseous pollutants removal, while Section 2.3 deals with polythiophene and other conductive polymers and their derivatives as adsorbents for environmental remediation applications.

### 2.1. PANI and PANI-Based Composite Adsorbents for the Removal of Heavy Metal Ions (HMIs)

The various polyaniline-based adsorbents are discussed here, which as divided into HMIs and OPs. Removal of various heavy metals ions and organic dyes by polyaniline and its derivatives is presented in a recent review by [25]. Heavy metal ions (HMIs) removed by polyaniline-based nanocomposites include the removal of Pb (II) ions by polyaniline-modified multiwalled carbon nanotubes [26] under ambient conditions. It was concluded that due to the high affinity of amine and imine functional groups of PANI toward Pb (II) ions, the PANI modification significantly improved the adsorption capacity. Polyaniline synthesized on jute fiber surfaces for Cr (VI) removal was reported by [27]. At the optimum experimental conditions (pH 3 and temperature of 20 °C), a maximum monolayer adsorption capacity of 62.9 mg/g was observed. It was reported that the total chromium adsorption decreased with increasing temperature, suggesting an exothermic nature of the chromium adsorption process. Additionally, utilization of polyaniline-coated polyacrylonitrile fiber mats for Cr (VI) removal was reported by [28], which concluded that PANI/PAN composite exhibited superior removal capabilities for Cr (VI). The maximum adsorption capacity was observed to increase with increasing temperature, which is suggestive of the endothermic nature of the adsorption process. In another research, Cr (VI) removal by polyaniline-coated carbon fiber fabric, cellulose–polyaniline composites was reported by [29]. They concluded that the introduction of PANI to their substrates improved both the adsorption rates and adsorption capacities. They reported that since the pseudo-second-order kinetic model fits well the experimental data, the adsorption process is physical adsorption in nature. The use of PANI-based adsorbents for the removal of Cr (VI) was extensively reported by [30].

The use of sodium alginate–polyaniline nanofibers for Cr (VI) ions adsorption was reported by [31] and observed that electrostatic interactions between the sodium alginate–polyaniline nanofibers and Cr (VI) were involved in the adsorption process. A maximum adsorption capacity of 73.34 mg/g was deduced from the Langmuir isotherm plots at 30 °C and the pseudo-second-order model fitted well the experimental data. Furthermore, the removal of Hg (II) from aqueous media was reported by [32], using polyaniline nanocomposite coated on rice husk ash. The removal of resorcinol from an aqueous solution was reported by [33], using SBA-15/polyaniline mesoporous composites. Polyaniline nanofibers assembled on alginate microspheres are used for the removal of Pb (II) and Cu (II) ions from aqueous media [34]. Cadmium metal ions Cd (II) removal from aqueous solution was reported by [35], using polyaniline-coated sawdust and Pb (II) and Cd (II) ions using polyaniline grafted chitosan [36]. The use of PANI–clay hybrid materials for the removal of Cu (II) was reported by [37]. Additionally, removal of arsenic (III) ions by magnetic polyaniline doped-strontium titanium nanocomposites adsorbent were reported by [38]. The maximum adsorption capacity was found to be 67.11 mg/g from the Langmuir isotherm model, and the nature of the adsorption process of arsenic (III) ions was found to be exothermic and physisorption deduced from the thermodynamic studies. The various adsorption mechanisms proposed for heavy metal ions removal are shown in Figure 2a,b, while a list of PANI and PANI-based adsorbents dealing with heavy metals ions removal under different experimental conditions are tabulated in Table 2.

### 2.2. PANI and PANI-Based Composite Adsorbents for the Removal of Organic Dyes and Other Organic Pollutants from Aqueous Environments

Polyaniline (PANI) and its derivatives possess significant detoxification characteristics and have been exclusively reported in the literature. The utilization of polyaniline nanoparticles as an adsorbent material for the removal of MB dye was reported by [45]. They reported that the synthesized material can be used as an efficient adsorbent for MB removal from water. The pseudo-second-order kinetic model was reported to fit well the experimental data. Moreover, they reported that using PANI nanoparticles and conventional PANI. The researchers concluded that PANI nanoparticles are more efficient adsorbents for MB, compared with conventional PANI powdered adsorbents, and found that the pseudo-second-order kinetic model best described the experimental results.

In another research, the removal of reactive black 5 (RB-5) and reactive violet 4 (RV-4) dyes was reported by [46], using PANI–starch nanocomposite. They reported high removal efficiencies of 99% and 98% for RB-5 and RV-4, respectively. Toth isotherm model was reported to best fit the equilibrium experimental data of both dyes. The main adsorption interactions between the adsorbent and the MB molecules reported are due to the availability of surplus hydrogen groups from the starch material. The ionic interactions were further confirmed by FTIR and desorption studies. Cationic dye, methylene blue (MB), and anionic dye, and Procion red (PR) removal from aqueous solutions using acid and base treated polyaniline were reported by [47]. Their experimental data were represented by the Langmuir isotherm equilibrium model. Furthermore, they stated that the cationic dye was mainly removed by the base-treated PANI, while the anionic dye was predominantly removed by the acid-treated PANI. The removal of an anionic dye, Rose Bengal (RB) aand a cationic dye, methylene blue (MB), from aqueous solutions using polyacid-doped polyaniline was reported by [48]. They reported maximum adsorption capacities of 440.00 mg/g for RB and 466.5 mg/g for MB. The adsorption mechanism reported involved π–π interactions and electrostatic interactions. The pseudo-second-order kinetic model and Langmuir isotherm equilibrium model followed the adsorption experimental data. In addition, the utilization of polyaniline in the form of nanoporous hyper-crosslinked polyaniline (HCPANI) for the removal of two types of dyes—cationic dye, crystal violet (CV) and anionic dye, methyl orange (MO)—was reported by [49]. The reported maximum adsorption capacities for the two dyes were 245 mg/g and 220 mg/g for CV and MO, respectively. The main proposed interactions involved in the adsorption process of CV and MO dyes include π–π interactions, hydrogen bonging, acid–Lewis-based interactions, as well as electrostatic interactions.

Adsorption of various organic pollutants from aqueous and non-aqueous sources was reported by [50], using highly porous carbons obtained by the pyrolysis of polyaniline conductive polymer. High adsorption capacities were reported for various pollutants investigated. The reported mechanisms are mainly due to hydrogen bonding, in addition to π–π interactions and/or hydrophobic interactions between the adsorbent (polyaniline-derived carbons, PDC-700, polyaniline-derived carbon activated at 700 ºC) and the adsorbates investigated. The adsorbent was tested for the successful removal of diethyl phthalate from an aqueous solution and for the removal of 4,6-dimethyldibenzothiophene from a model fuel. Various adsorption interactions between the adsorbent and adsorbate are reported including hydrophobic interactions, π–π interactions, electrostatic interactions, hydrogen bonding, and acid–base interactions. Again, the very high adsorption capacity of the prepared adsorbent material for different organic pollutants is mostly due to the high porosity, high specific surface area, and presence of various functional groups.

Polyaniline-based nanocomposites are being increasingly used for the removal of organic pollutants and organic dyes. These include the work of [51] for the removal of MB dye from wastewater using polyaniline zirconium (VI) silicophosphate nanocomposites. The nature of the adsorption behavior of this composite was found to be spontaneous. The adsorption of MB on the reported adsorbent was reported to follow a second-order kinetic model, and the experimental data were best fitted by the Freundlich isotherm model. A maximum adsorption capacity of 12 mg/g was deduced from the Langmuir isotherm model fitting to the experimental data. Furthermore, methyl orange removal by polyaniline/MWCNTs/Fe_3_O_4_ composites [52]. Another important paper concerned with the removaland decolorization, of Remazol effluent includes the work of [53], who reported the use of bacterial extracellular polysaccharides–polyaniline composites. Adsorption of brilliant green (BG) was reported by [54], using polyaniline/silver nanocomposites. In another paper, the removal of basic blue dye was reported [55], using polyaniline/magnetite (Fe_3_O_4_) nanocomposites. Removal of another organic material such as tetracycline hydrochloride was reported by [56], using polypyrrole coated iron-doped titania-based hydrogel. Utilization of polyaniline-based adsorbents for the removal of dyes from water and wastewater was reported in detail in the review of [57]; some of the proposed mechanisms for MB removal by PANI are shown in Figure 3. Decolorization of Acid Blue 29 (an azo dye) was reported by [58] by utilizing PANI–ZnO–ZrO_2_ composite as a photocatalyst in UV photocatalytic reactor. They reported that the composite showed better decolorization of the dye, compared with PANI alone.

In another research article, Rhodamine G6 (Rh-G6) was photocatalytically degraded by polyaniline–zinc sulfide (PANI–ZnS) nanocomposite with a removal efficiency of about 80%, as reported by [59]. Photodegradation of MB and MG dyes by PANI–ZnO nanocomposite was reported by [60], in which the degradation was conducted under natural sunlight and under UV radiation. They reported high removal efficiencies for both dyes under natural sunlight exposure of 5 h. In addition, the authors of [61] studied the removal of MB from an aqueous solution using PANI–ZrO_2_ nanocomposite. The effects of various process parameters on the adsorption characteristics were reported. The dye removal efficiency was found to increase with increasing contact time and operating temperature. The reported maximum adsorption capacity for the PANI-modified ZrO_2_ was found to be 77.55 mg/g. Further, photodegradation of MB was reported by [62], using polyaniline–zirconium silicophosphate (PANI–ZSP). The reported nanocomposite initially adsorbed the MB molecules on its surface active sites and then degraded the MB upon exposure to visible light. After two hours of exposure to visible light, a degradation efficiency of 82% was attained.

Organic pollutants other than organic dyes also require treatment before being discharged into the aquatic environment. The use of PANI and its derivatives have been extensively used as adsorbents for organic pollutants removal. For example, the removal of tannic acid from wastewater was reported by [63], using synthesized polyaniline. They reported that the Langmuir isotherm and pseudo-second-order kinetic models fitted well with the experimental data. A very high maximum adsorption capacity was observed at a high ionic strength of 2 moles per liter of NaCl solution. The nature of adsorption of tannic acid over the synthesized PANI was mainly suggested to be chemisorptions. The initial solution pH was reported to have a significant effect on the TA adsorption on PANI. The adsorbent–adsorbate interactions were mainly due to hydrogen bonding, electrostatic attraction, π–π interactions, as well as a weak Van der Waals force. A more comprehensive list of various ODs and OPs removal by PANI and PANI-based adsorbents under various experimental conditions is provided in Table 3 below.

#### PANI and PANI-Based Composite Adsorbents for the Removal of Gaseous Pollutants

Utilization of PANI-derived porous and nitrogen-doped carbon materials with very high specific surface area for CO_2_ uptake was reported by [81]. This study focused on the adsorption of various gases such as N, CO_2_, and CH_4_ over the prepared material and reported selective adsorption of CO_2_, compared with N and CH_4_, and relatively high capture capacity for the synthesized adsorbent for CO_2_ uptake. The nature of adsorption was reported to be physisorption or weak chemisorption. Removal of ammonia gas by PANI–TiO_2_ as photocatalyst was reported by [82] under visible light and under UV radiation. They reported that the removal efficiency decreased as the reaction time increased.

Moreover, CO_2_ reduction to alcohol by polyaniline film was reported by [83]; their proposed reaction mechanisms are presented in Figure 4. Removal of various volatile organic compounds (VOCs) by various forms of polyaniline was reported by [84]. They reported that the main mechanisms which are at play in the removal of VOCs are the π–π interactions between PANI backbone and the unsaturated hydrocarbons, which resulted in higher removal of unsaturated (C=C) bonds present in the target analytes. As for the saturated hydrocarbon-based VOCs, the main interactions are weak hydrogen bonding and weak Van der walls forces between PANI and the saturated molecules owing to the lack of available π electrons. Overall, the type of PANI (EB or ES), surface area, morphology, and the type of doping agent (dopant) can significantly affect the VOC–PANI interactions and the removal performance. Adsorption of a flue gas NO_2_ by polyaniline–clay nanocomposite was reported by [85]. They reportedly prepared polyaniline composites with three different clays—namely, attapulgite (ATP), vermiculite (VEM), and diatomite (DIM), and concluded that the PANI–ATP composite revealed the high adsorption capacity for NO_2_ removal. Surface morphologies of some PANI-based adsorbents are presented in Figure 5.

### 2.3. Polypyrrole-Based Nanocomposites as Adsorbents

#### 2.3.1. HMIs Removal by Ppy and Ppy-Based Composite Adsorbents

Hexavalent chromium removal by exfoliated polypyrrole–organically modified MMT clay nanocomposites was reported by [94]. The researcher reported that the kinetic data fitted well the PSO kinetic model, while the equilibrium data were best fitted by the Langmuir model. Additionally, they concluded that an increase in maximum adsorption capacity from 112 to 209 mg/g at temperatures 19 to 45 °C. Polypyrrole-functionalized chitin was used for Cr (VI) removal by [95], who reported that the Freundlich isotherm model fitted well the experimental data. The reported maximum adsorption capacities ranged from approximately 29 to 35 at 30–50 °C temperature. The adsorption process was reported to be spontaneous and endothermic. Cr (VI) by polypyrrole-wrapped MWCNTs nanocomposites was reported by [96], who found a maximum adsorption capacity of 294 mg/g. They also reported good fitting of the experimental data by the Langmuir model and the spontaneous and endothermic nature of adsorption for Cr (VI) removal. Threonine-doped polypyrrole nanocomposites for Cr (VI) removal were reported by [97]. Additionally, the removal of Cr (VI) from wastewater was reported by [98], using polypyrrole/2,5-diaminobenzene sulfonic acid composite and by glycine doped polypyrrole composite [99].

Uranium (VI) ions removal by polypyrrole was investigated by [100] in a batch system and reported that the Freundlich isotherm model was in good agreement with their experimental data. A maximum adsorption capacity of 87.72 mg/g was deduced from the Langmuir isotherm model and the pseudo-second-order kinetic model showed a better correlation. They also reported the endothermic and spontaneous nature of the adsorption process for uranium (VI) ions by exploring the thermodynamic data of their work. The use of polypyrrole for the removal of copper ions from aqueous solutions was reported by [101], who stated that the Langmuir isotherm model fitted well the experimental data. They also concluded that the available amine functional groups for ion exchange in polypyrrole made it a good adsorbent. The removal of Cr (VI) from aqueous solutions using bamboo-like polypyrrole nanotubes was reported by [102], who observed higher adsorption performance for their synthesized nanotubes, compared with conventional polypyrrole adsorbents for Cr (VI) removal.

In another paper, removal of Cr (VI) from aqueous solution was reported [103], using polypyrrole/monodisperse latex spheres and also by using polypyrrole/calcium rectorite composite [104]. Further, the removal of Cr (VI) and Cu (II) metal ions from aqueous media was reported by [105], using polypyrrole–maghemite magnetic nanocomposites. The removal of another heavy metal Hg (II) ions by polypyrrole-functionalized CoFe_2_O_4_@SiO_2_ was reported by [106]. Adsorption of heavy metal Pb (II) ions in polypyrrole–bentonite nanocomposites [107] and in polypyrrole–Fe_3_O_4_ nanosized magnetic adsorbents were also reported [108]. The removal of cadmium Cd (II) ions from wastewater was reported by [109], using polypyrrole–TiO_2_ nanocomposites. A detailed review on the utilization bio-composites coated by polypyrrole was reported by [110], while another study pointed to microbial fuel cells, coupled with Fenton oxidation [111], for the removal of heavy metal ions from wastewater. Polypyrrole-coated sawdust of dryobalanops aromatic for the adsorption of cadmium-109 isotopes was reported by [112]. The removal of arsenic from wastewater was reported by [113], using polypyrrole composites with bentonite and activated carbon. Some of the proposed adsorption mechanisms for Cr (VI) removal by polypyrrole-modified magnetic nanocomposites are demonstrated in Figure 6. In addition, some HMIs removal by polypyrrole-modified adsorbents are listed in Table 4.

#### 2.3.2. Removal of Organic Pollutants and Organic Dyes by Ppy and Ppy-Based Composite Adsorbents

Different organic pollutants including organic dyes may present significant threats to the environment and to human health. However, their removal and remediation from wastewater were reported by numerous researchers. These include, for example, the removal of MB from aqueous solution using polypyrrole-coated cotton fabrics [118] and polypyrrole–TiO_2_ composites [119], as well as the removal of Congo red by molecularly imprinted polypyrrole-coated magnetic TiO_2_ nanoparticles. Removal of naphthol green B from aqueous solution was reported by [120] using polypyrrole/Attapulgite composites. The removal of acidic dye namely Congo red by various polypyrrole-based composite adsorbents was reported by [121]. The adsorption behavior of various anionic and cationic organic dyes was reported by [122] by using polypyrrole–SBA-15 nanocomposites. The removal of another dye, atrazine, by nylon–polypyrrole core shells nanofibers mat was reported by [123]. A well-detailed review for the utilization of polypyrrole-based composite was reported by [124] for the removal of acid dyes. Polypyrrole nanofibers with hierarchical structure for the removal of acid red G (azo dye) were reported by [125]. They reported a maximum adsorption capacity of 121.95 mg/g for their investigated dye. Further, Ppy–MWCNT nanocomposite was used as an adsorbent for the removal of a non-steroid anti-inflammatory drug (potassium diclofenac) from an aqueous solution [126]. They reported that the modification of MWCNT by Ppy has significantly improved the maximum adsorption capacity and that the thermodynamic parameters suggested endothermic and favorable adsorption. Further, polypyrrole-based adsorbent—namely, polypyrrole-functionalized *Calotropis gigantea* fibers are being successfully used for the removal of three fluoroquinolone antibiotics from wastewater, as reported by [127]. The prepared adsorbent exhibited superior adsorption capacities for the investigated antibiotics. Further, they reported that the main adsorption mechanism may be hydrophobic interactions, electrostatic interactions, ion exchange, π–π interactions, and hydrogen bonding. Figure 7 shows some of the proposed adsorption mechanisms for organic dye removal by polypyrrole. Adsorption of another organic compound, 4-nitrophenol, by polypyrrole–bentonite clay nanocomposite was reported by [128]. A maximum adsorption capacity of 96 mg/g of adsorbent was reportedly deduced from the Langmuir isotherm model. The thermodynamic parameters suggested an exothermic adsorption process. In another important work, the simultaneous removal of various polycarboxy–benzoic acids by polypyrrole–nut shells of argan (Ppy–NA) was reported by [129]. They reported relatively high adsorption capacity of the prepared adsorbent material for all acids. They reported that the adsorption process is spontaneous and endothermic in nature. Furthermore, the removal of some organic dyes by conductive polymers is listed in Table 5.

#### 2.3.3. Gaseous Pollutants Removal by Ppy and Ppy-Based Composite Adsorbents

Removal of gaseous pollutants by Ppy and Ppy-based composite adsorbents is relatively scarce in the literature. However, adsorption of CO_2_ on porous rodlike polypyrrole structure was reported by [136], who concluded that the maximum CO_2_ uptake was 173.885 mL/g at 195K for Ppy synthesized without any surfactant.

### 2.4. Polythiophene and Other Conductive Polymer Nanocomposites as Adsorbents

Polythiophenes and their derivatives are rarely used as adsorbents for the removal of potentially hazardous pollutants. However, the environmental remediation of Cr (III) using polythiophenes-based adsorbent with a maximum adsorption capacity of 85.79 mg/g was reported by [137]. The removal of organic dye methylene blue using polythiophene-modified adsorbents was reported by [138], with relatively high adsorption capacities. The experimental data were reported to be best described by the pseudo-second-order kinetic model, while the thermodynamic parameters reportedly suggested a spontaneous and endothermic nature of the adsorption process. As a photocatalyst, polythiophene-based nanocomposites were reported for the removal of MB and MO by [139], under visible LED and natural sunlight. The prepared nanocomposite was reported to have excellent photocatalytic and adsorption activity, while the prepared material can be easily separated from the solution after equilibrium by the use of an external magnet. Photodegradation of MB was also reported by [140], using polythiophene-doped SrTiO_3_ nanocomposite, and reported better catalytic activity for the prepared nanocomposite, compared with the starting materials, i.e., polythiophene and SrTiO_3_. In another study, removal of arsenic ions by sawdust, modified by the three well-known CPs—namely, PANI–SD, Ppy–SD, and PTh–SD, was reported by [141]. The study concluded that PTh–SD adsorbent showed the highest adsorption capacity, compared with unmodified SD, PANI–SD, and Ppy–SD samples. The exact adsorption mechanism was clearly stated, but three possible reactions were proposed that include chemical oxidation, anion exchange, and chelation processes. Furthermore, it was reported by [142] that Cd (II) can be efficiently removed by polythiophene nanocomposites. Likewise, the use of polythiophene as an adsorbent material was reported by [143] for the removal of toxic As (III). In the reported adsorption process, the As ions become attached to π electrons at the backbone of the polymer, leading to much stronger interactions between S and As atoms.

#### Pollutants Removal by Combined Conductive Polymers

The use of combined conductive polymers for pollutants removal has been reported by various researchers. For example, PANI–Ppy nanofibers for the removal of Cr (VI) were reported by [115]. Similarly, the use of PANI–Ppy copolymer nanofibers for the removal of cobalt ions Co (II) from aqueous solutions was reported by [144]. They emphasized the positive role of temperature on the adsorption process and reported 99.68% removal efficiency for a 100 mg/L Co (II) concentration at the optimum operating conditions. Similarly, the removal of Congo red (CR) from aqueous solutions using PANI–Ppy nanofibers was reported by [131] in a batch adsorption model, indicating higher removal efficiencies for CR at low solution pH. They also reported good fitting of the Langmuir isotherm equilibrium model and pseudo-second-order kinetic model to their experimental data. A higher adsorption capacity was observed for PANI nanofibers (270.27 mg/g), compared with Ppy nanofibers (222.22 mg/g). Further utilization of conductive polymers as adsorbents for nitrates from wastewater was reported by [145], using polyaniline and polypyrrole as adsorbents. They reported that the Langmuir isotherm model fitted well the experimental data and that the adsorption process followed the pseudo-second-order kinetic model. The nature of the adsorption of nitrates on PANI and Ppy was spontaneous. However, higher adsorption capacities for nitrates were observed by PANI, compared with Ppy. The removal of Congo red was reported for PANI and Ppy adsorbents in another paper by [146], who observed that removal efficiencies increased with increasing contact time and adsorbent dosage. They reported good fitting of the Langmuir equilibrium isotherm and pseudo-second-order kinetic models with their experimental data. In another research article, methylene blue (MB) removal using polyaniline and polypyrrole macro-nanoparticles was reported [147]. The reported maximum adsorption capacity for the synthesized nanoparticles was 19.2 mg/g of MB/g of polymer. Additionally, a detailed review on the utilization of conducting polymers as adsorbents for the removal of textile dyes was reported by [57]. Various surface morphologies possessed by polypyrrole-modified adsorbents are presented in Figure 8 below.

## 3. Conclusions

Conductive polymers and their composites are known to be efficient adsorbents for various types of pollutants and contaminants. This is mostly due to their interesting redox characteristics and the presence of N, S, P, and O elements in their chemical structure. The most predominant reported mechanisms for organic dyes removal are π–π interactions, hydrogen bonding, hydrophobic interactions, acid–base interactions, and electrostatic interactions. As for the heavy metal ions removal, the most common mechanisms are electrostatic attraction, ion exchange, chelation, and reduction. Among the heavy metals, removal of Cr (VI) is the most widely studied contaminant, while among the organic dyes, methylene blue is reportedly the most widely studied pollutant. Overall, it may be concluded that modification by conductive polymers of various types of potential adsorbent materials leads to significant improvements in the adsorption rates and maximum adsorption capacities of the unmodified adsorbents. Among the conductive polymers, polyaniline and polypyrrole have been extensively studied as potential adsorption enhancers (leading to significantly high maximum adsorption capacities), compared with polythiophenes. Hence, polythiophenes and their derivates may present opportunities for further exploration and research.

## Figures and Tables

**Figure 1 polymers-13-03810-f001:**
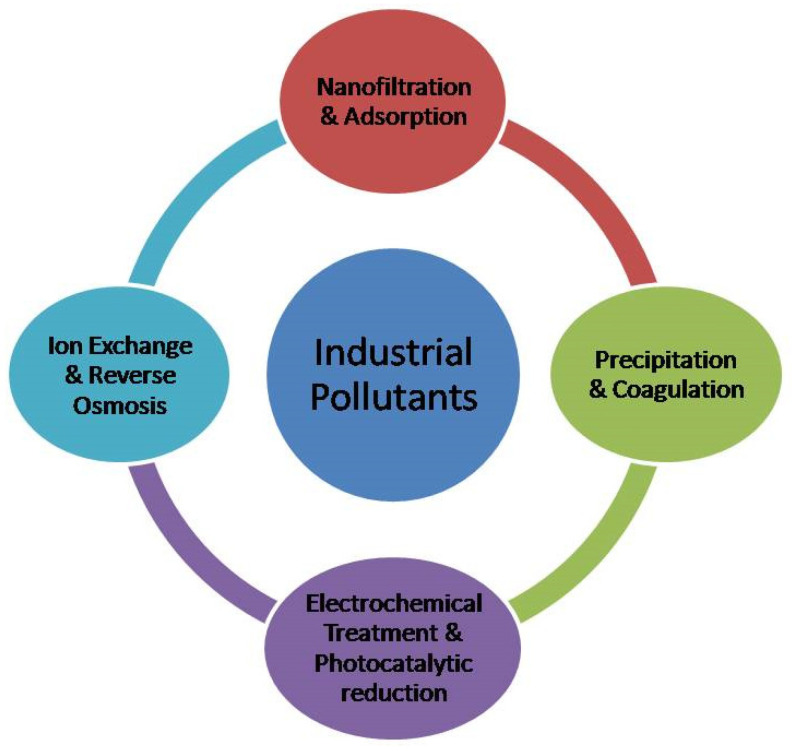
Various available treatment techniques for industrial pollutants.

**Figure 2 polymers-13-03810-f002:**
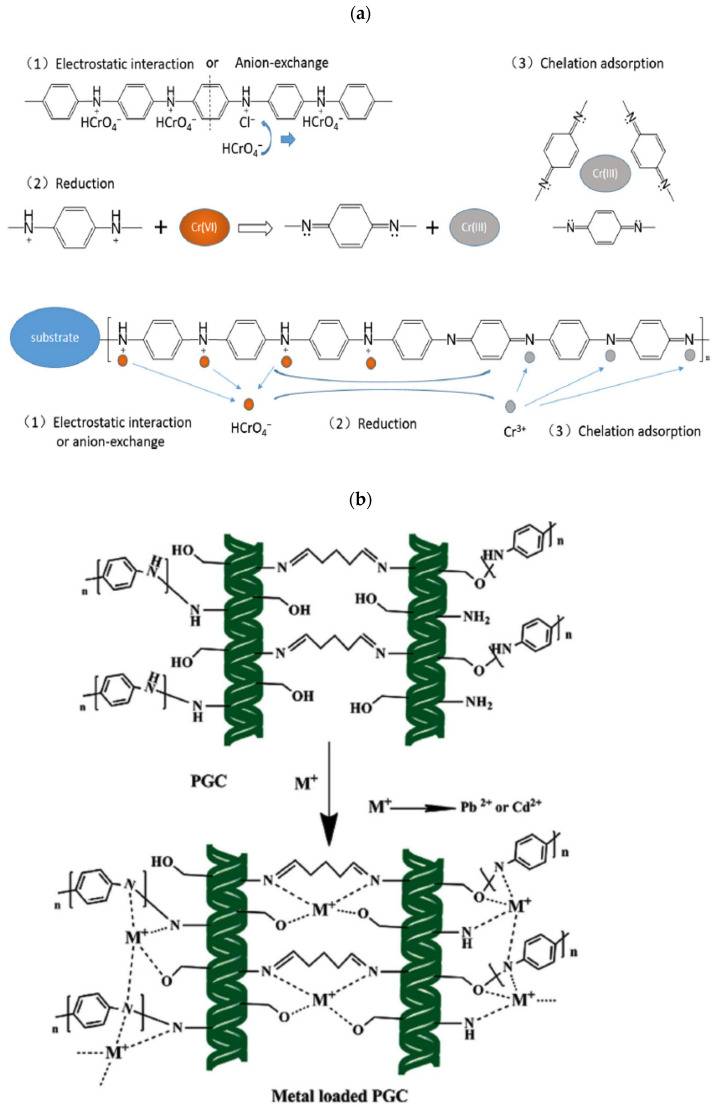
(**a**) Mechanism of chromium removal by PANI and PANI-based composites. Reprinted with permission from Ref. [30]. Copyright 2018 Springer Nature. (**b**) Proposed mechanism for the adsorption of Pb and Cd onto PANIGCS. Reprinted with permission from Ref. [25]. Copyright 2015 Elsevier.

**Figure 3 polymers-13-03810-f003:**
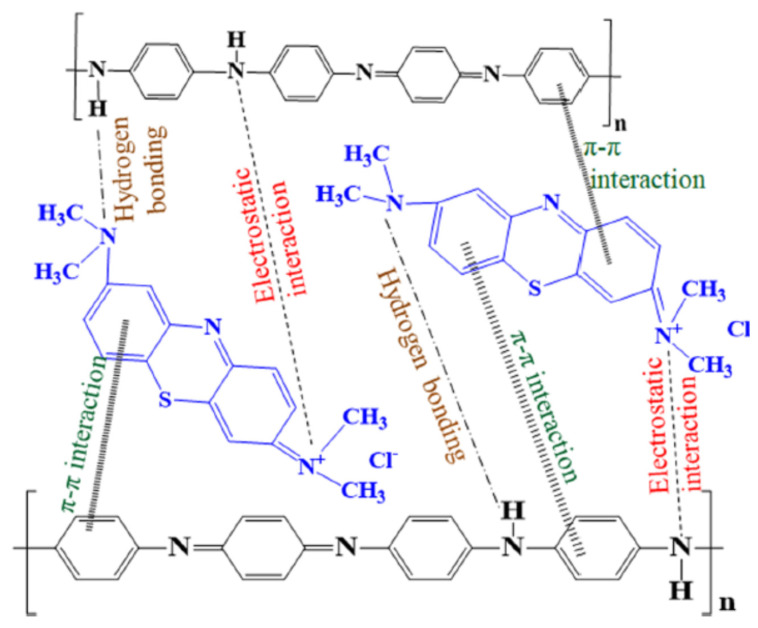
Different interactions proposed for the mechanism of methylene blue adsorption on polyaniline. Reprinted with permission from Ref. [57]. Copyright 2019 Springer Nature.

**Figure 4 polymers-13-03810-f004:**
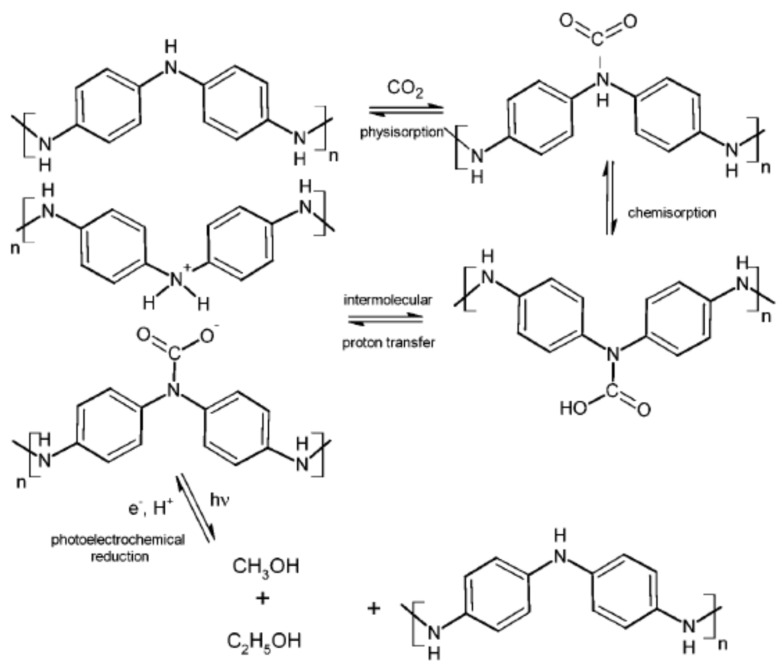
Possible reaction pathway for CO_2_ adsorption and photoelectroreduction on PANI. Reproduced from Ref. [83]. Copyright 2016 Royal Society of Chemistry.

**Figure 5 polymers-13-03810-f005:**
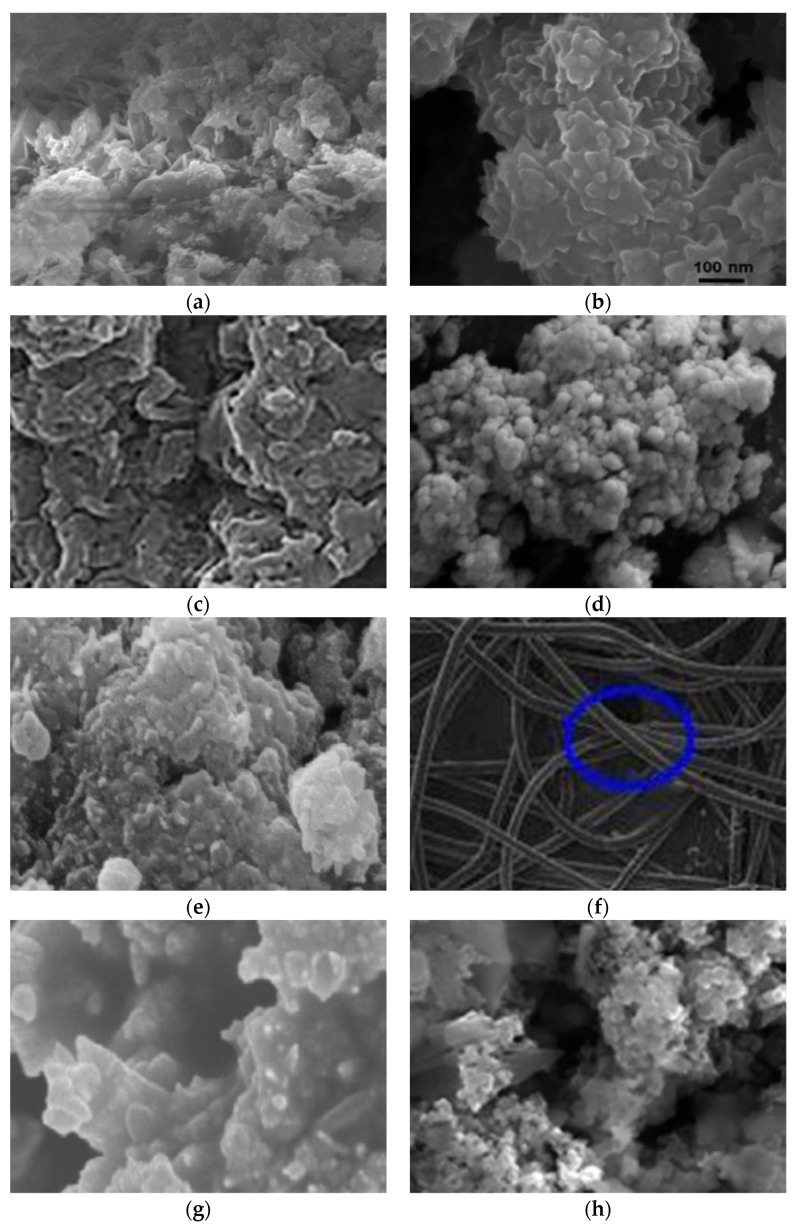
SEM images of some representative PANI and PANI-based polymer composites. (**a**) Reprinted with permission from Ref. [86]. Copyright 2017 Elsevier. (**b**) Reprinted with permission from Ref. [87]. Copyright 2019 Elsevier. (**c**) Reprinted with permission from Ref. [88]. Copyright 2018 Elsevier. (**d**) Reprinted with permission from Ref. [89]. Copyright 2018 Elsevier. (**e**) Reprinted with permission from Ref. [90]. Copyright 2010 Elsevier. (**f**) Reprinted with permission from Ref. [91]. Copyright 2013 Elsevier. (**g**) Reprinted with permission from Ref. [92]. Copyright 2014 Elsevier. (**h**) Reprinted with permission from Ref. [93]. Copyright 2013 Wiley Periodicals, Inc.

**Figure 6 polymers-13-03810-f006:**
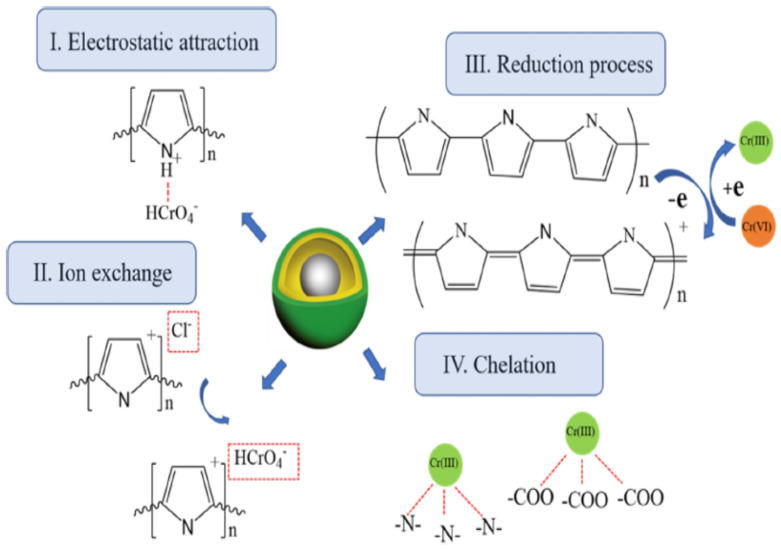
Possible adsorption mechanism of Cr (VI) removal by magnetic-UiO66-Ppy. Reprinted with permission from Ref. [114]. Copyright 2021 Springer Nature.

**Figure 7 polymers-13-03810-f007:**
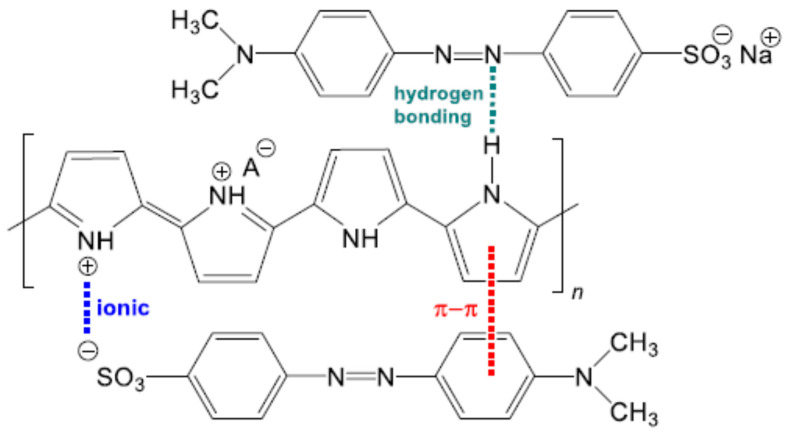
π–π interactions and hydrogen bonding between organic dye methyl orange and Ppy. Reprinted with permission from Ref. [130]. Copyright 2019 Springer Nature.

**Figure 8 polymers-13-03810-f008:**
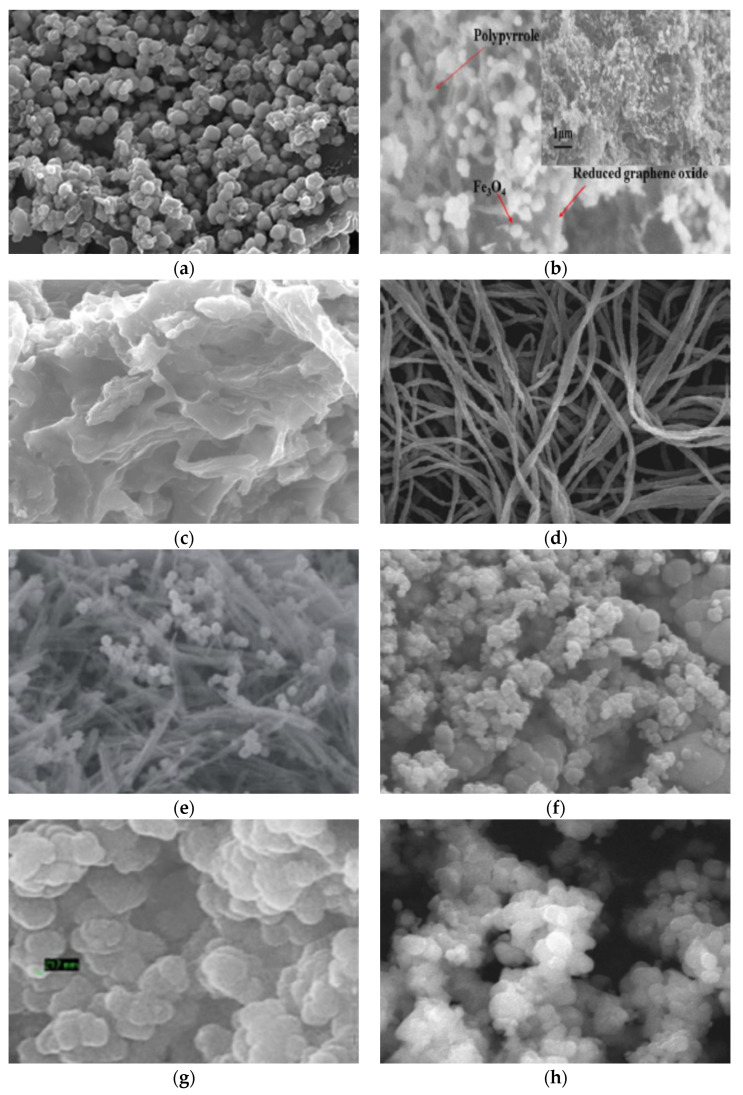
Some representative SEM images of Ppy and Ppy-based composite materials. (**a**) Magnetic Fe_3_O_4_@Arg-Ppy nanocomposite. Reprinted with permission from Ref. [148]. Copyright 2018 Elsevier. (**b**) Ppy-Fe_3_O_4_/rGO composite. Reprinted with permission from Ref. [149]. Copyright 2014 Elsevier. (**c**) Ppy-Nutshell of Argan composite. Reprinted with permission from Ref. [129]. Copyright 2016 Elsevier. (**d**) Ppy-Bacterial Cellulose Fiber composite. Reprinted with permission from Ref. [150]. Copyright 2021 Springer Nature. (**e**) Ppy-mixed oxide nanocomposite. Reproduced from Ref. [151]. Copyright 2018 Royal Society of Chemistry. (**f**) Ppy-TiO_2_ nanocomposite. Reprinted with permission from Ref. [152]. Copyright 2012 Elsevier. (**g**) Ppy-Magnetic Corncomb Biochar composite. Reprinted with permission from Ref. [153]. Copyright 2018 Elsevier. (**h**) Fe_3_O_4_-TiO_2_-Ppy nanocomposite. Reprinted with permission from Ref. [154]. Copyright 2016 Springer Nature.

**Table 1 polymers-13-03810-t001:** Some conductive polymers and their chemical structures, Adapted from Ref. [6].

Polymer	Abbreviation	Structure
Polythiophene	PTh	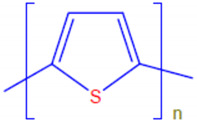
Polypyrrole	Ppy	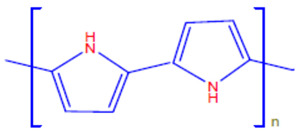
Polyaniline	PANI	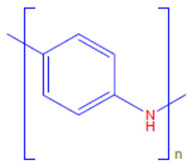
Polyacetylene	PA	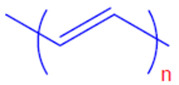
Functionalized Polyacetylene	f-PA	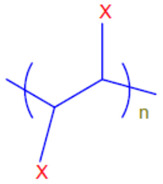
Poly(phylene vinylene)	PPv	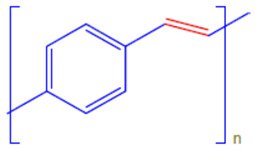
Poly(p-phenylene)	PP	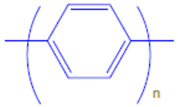

**Table 2 polymers-13-03810-t002:** PANI-based adsorbents for the removal of HMIs PANI-based adsorbents for the removal of HMIs.

Adsorbent	Adsorbate	pH	Temperature (°C)	q_max_ (mg/g)	Ref.
PANI-JF	Cr (VI)	3	40	62.89	[27]
PANI/CA composite	Cu^+2^, Pb^+2^	6	NA	67.95 251.44	[34]
PANI-PGC	Pb^+2^ Cd^+2^	6	30	16.07 14.33	[36]
PANI/Clay	Cu^+2^	6	25	22.77	[37]
PANI-PS	Pb^+2^	2-6	NA	NA	[39]
PANI-Chi composite	Cr (VI)	3	45	1.01	[40]
PANI-BC mat	Cr (VI)	1-2	NA	NA	[41]
P-PANi-MMT	Cu^+2^	5	NA	87	[42]
PANI-PVDF-HFP nanofibrous	Cr (VI)	4.5	NA	15.08	[43]
PANI and PANI-G10	Cr (VI)	6.5	30	136, 192	[44]

**Table 3 polymers-13-03810-t003:** PANI-based adsorbents for the removal of ODs and other organic pollutants.

Adsorbent	Adsorbate	pH	Temperature (°C)	q_max_ (mg/g)	Ref.
PANI-SBA-15	resorcinol	3	25	128	[33]
HCPANI	MO, CV	3, 11	27	220, 245	[49]
PANI-ZSP	MB	1	85	12	[51]
PANI-MWCNTs-Fe_3_O_4_ magnetic composite	MO CR	4	Room temp.	446.25, 417.38	[52]
Fe_3_O_4_ and PANI-Fe_3_O_4_	BB-3	8.5, 12, 10	30	8.5, 6, 9	[55]
PANI-AC	MO	6.5	25	285	[64]
PANI-AC	Direct Red 23	3.0	45	109.89	[65]
PANI and PANI/AL	DG	1	20	0.911, 8.13	[66]
PANI-Chi	CR, CBB, RBBR	3	26	322.58, 357.14, 303.03	[67]
PANI-MMT-Fe_3_O_4_	MB	6.3	Room temp.	184.48	[68]
PANI/CPL	MO	4	Room temp.	333.33	[69]
PANI, Fe_3_O_4_, and PANI-Fe_3_O_4_	AB-40	3, 6, 6	30	130.5, 264.9, 216.9	[70]
PANI-Fe_3_O_4_	MG	7	25	4.82	[71]
PANI-HGL	MB	6.5	45	71.2	[72]
PANI-LC	RB-5	2.0	Room Temp.	312	[73]
PANI-LC	CR	4.29	45	1672.5	[74]
PANI-NFs/SD	ARG	2.0	35	212.97	[75]
PANI-FeCl_3_	RB-5	6	45	434.7	[76]
PANI-NiFe_2_O_4_	MG	7	N/A	4.09	[77]
PANI-NiFe_2_O_4_	ARS	4 8.6	30	186	[78]
PANI-Ny-6	MO	1	N/A	370	[79]
PANI-ZnFe_2_O_4_	RH-B	2	Room tem.	229	[80]

**Table 4 polymers-13-03810-t004:** Ppy-based adsorbents for the removal of HMIs.

Adsorbent	Adsorbate	pH	Temperature (°C)	q_max_ (mg/g)	Ref.
Ppy-PANI	Cr (VI)	2	25	227	[115]
Ppy-oMMT NC	Cr (VI)	2	25	209.6	[94]
Ppy-Chitin	Cr (VI)	2	50	35.22	[95]
Ppy/Fe_3_O_4_ and Ppy/oMWCNTs NC	Cr (VI)	2	45, 25	243.9, 294	[96]
Ppy/DABSA	Cr (VI)	2	25	303	[98]
Ppy-gly	Cr (VI)	2	45	232.55	[99]
Ppy/MLS	Cr (VI)	2	25	343.64	[103]
Ppy/Ca-REC	Cr (VI)	1.5	45	833.33	[104]
Ppy-Fe_3_O_4_	Hg_2+_	2.5	55	173.16	[108]
Ppy–BOFS NC	phosphate	2	45	9.13	[116]
Ppy-GSi NC	Cr (VI)	2	25	429.2	[117]

**Table 5 polymers-13-03810-t005:** Ppy-based adsorbents for the removal of ODs.

Adsorbent	Adsorbate	pH	Temperature (°C)	q_max_ (mg/g)	Ref.
Ppy-PANI NF	CR	4	35	270.27	[131]
Ppy NF	ARG	2	25	121.95	[125]
Ppy-CF	MB	12	Room Temp.	6.0	[118]
Ppy/TiO_2_	MB	13	35	298.50	[119]
Ppy-Attapulgite-ZVI	NG-B	2	45	253.9	[120]
Ppy-Chi-LS	CR	2	50	30.12	[121]
Ppy-SBA-15 NC	MB MO	4.5, 6.5	20	58.82, 41.66	[122]
Ppy-PA6 NFM	atrazine	7	70	14.8	[123]
Ppy-SD	AO-10	3	45	256.41	[124]
Ppy-BNT NC	4-nitrophenol	N/A	25	96.15	[128]
Ppy-α Cellulose	RR-120	2	25	96.1	[132]
Ppy-Chi-Fe_3_O_4_	AG-25	N/A	Room temp.	32.754	[133]
Ppy-CF	MB	10	25	3.30	[134]
Ppy/SD	MB	2	Room temp.	34.36	[135]

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
