# Peer review of "Conductive Polymers and Their Nanocomposites as Adsorbents in Environmental Applications"

_polymers, 2021, doi:10.3390/polym13213810_

Round 1

Reviewer 1 Report

The review of the review article “Conductive Polymers and their Nanocomposites as Adsorbents in Environmental Applications”

This is well written review. There are some technical details that would improve the quality of the article.

The whole manuscript – check the superscripts and subscripts in numbers of chemical formula, including Tables

Line 25 – typo: polyhtiophene (PTh) -> polythiophene

Line 57 – Figure 1: please increase the font in the outer four circles

Line 68 – Table 1 – the capitation of the table should be written above the table

Line 83 – is bold necessary in “Conductive polymers and their nanocomposites as adsorbent materials have”

Line 144 – Table 2: After Table 2 should be a dot instead of double dot. Please check the valence – it should be written in superscript (colon adsorbate)  

Line 269 – NO2 -> NO2

Line 277 – Figure 4: Figure 4 should be bold. After Figure 4 should be a dot, not a double dot. The figure 4 is hard to read, the letters are too small. I suggest increasing the font size

Line 280 – the table 4 is not a table, it is a picture. I suggest changing “Table 4” into the “Figure 5”. Also I suggest SiO2 -> SiO2

Line 327 – Figure 5: The Figure 5 should be in bold. After Figure 5 should be a dot.

Line 367 – after the Figure 6 are two dots. Please remove one.

Lines 401-403 – Please rewrite this sentence to be clearer “For example, the work of [109] who reported the use of PANI-Ppy nanofibers for the removal of Cr (VI), cobalt ions Co(II) from aqueous solutions using PANI-Ppy copolymer nanofibers [138].”

Line 426 – Table 7 should be a picture. I suggest changing “Table 7 into the Figure”

Reviewer 2 Report

Dear all,

Greetings

Please find enclosed my comments regarding paper

Referenced as: polymers-1442183

Titled: Conductive Polymers and their Nanocomposites as Adsorbents in Environmental Applications

 The authors have performed nice review about using conductive as environmental adsorbent, but this review can be accepted for publication in Polymers, after fixing all these recommendations (Minor Revisions)

1) Title: Ok

2) Abstract: please add the best conductive polymer in all the review (which one gives the high adsorbents activities)

3) Keywords: HMIs; ODs; GPs you can add these too

4) Comments:

  • chain can be a significant threat to the environment [X] and to the human health [Y]
  • environmental issues in the developing countries [X] industries such as power plants [X], petrochemical industries[Y], hydrogen and cement manufacturing plants [5], please add references and put each references near to it application
  • Table 1 not clear (like copy paste)
  • f-PA is considered as conductive polymer or not?
  • Table 2 splited between two pages
  • What about Hg heavy metal, if there is any study, please add it?
  • Table 3 splited between two pages?
  • After adsorption of dyes or organic pollutants by the CPs, it will be then their release desorption or degradation, please add some comments regarding this issue?
  • For gas like SO2, CH4 are there any work about it using the same concept
  • Table 6 splited between two pages
  • Table 7 splited between two pages (Table or Figures?)

5) Conclusion: ok, please in your opinion which is the best CPs, which can be used for the HMIs remediation, compare between polyaniline and polypyrrole?

6) References:

please update them add some of 2021 and 2022

 [1] Hu, X., et al., Frontiers of Environmental Science and Engineering, 2022, 16(4),48;

[2] Jadhav, P., et al., Environmental Research, 2022, 204,112043;

[3] Sathe, S.M., et al., Environmental Research, 2022, 204,112135

With regards
